# Nanoporous Iron Oxide/Carbon Composites through In-Situ Deposition of Prussian Blue Nanoparticles on Graphene Oxide Nanosheets and Subsequent Thermal Treatment for Supercapacitor Applications

**DOI:** 10.3390/nano9050776

**Published:** 2019-05-21

**Authors:** Alowasheeir Azhar, Yusuke Yamauchi, Abeer Enaiet Allah, Zeid A. Alothman, Ahmad Yacine Badjah, Mu. Naushad, Mohamed Habila, Saikh Wabaidur, Jie Wang, Mohamed Barakat Zakaria

**Affiliations:** 1International Center for Materials Nanoarchitectonics (WPI-MANA), National Institute for Materials Science (NIMS), 1-1 Namiki, Tsukuba, Ibaraki 305-0044, Japan; horiatalbher@hotmail.com (A.A.); abeer.abdelaal@science.bsu.edu.eg (A.E.A.); 2Key Laboratory of Eco-Chemical Engineering, College of Chemistry and Molecular Engineering, Qingdao University of Science and Technology, Qingdao 266042, China; 3School of Chemical Engineering and Australian Institute for Bioengineering and Nanotechnology (AIBN), The University of Queensland, Brisbane, QLD 4072, Australia; 4Department of Plant & Environmental New Resources, Kyung Hee University, 1732 Deogyeong-daero, Giheung-gu, Yongin-si, Gyeonggi-do 446-701, Korea; 5Chemistry Department, Faculty of Science, Beni-Suef University, Beni-Suef 62511, Egypt; 6Advanced Material Research Chair, Chemistry Department, College of Science, King Saud University, P.O. Box 2455, Riyadh 11451, Saudi Arabia; ybadjah@ksu.edu.sa (A.Y.B.); mnaushad@ksu.edu.sa (M.N.); mhabila@ksu.edu.sa (M.H.); tarabai22@yahoo.com.sg (S.W.); 7Department of Chemistry, Faculty of Science, Tanta University, Tanta, Gharbeya 31527, Egypt

**Keywords:** nanoporous materials, iron oxide, carbon composites, Prussian blue, supercapacitors

## Abstract

This work reports the successful preparation of nanoporous iron oxide/carbon composites through the in-situ growth of Prussian blue (PB) nanoparticles on the surface of graphene oxide (GO) nanosheets. The applied thermal treatment allows the conversion of PB nanoparticles into iron oxide (Fe_2_O_3_) nanoparticles. The resulting iron oxide/carbon composite exhibits higher specific capacitance at all scan rates than pure GO and Fe_2_O_3_ electrodes due to the synergistic contribution of electric double-layer capacitance from GO and pseudocapacitance from Fe_2_O_3_. Notably, even at a high current density of 20 A g^−1^, the iron oxide/carbon composite still shows a high capacitance retention of 51%, indicating that the hybrid structure provides a highly accessible path for diffusion of electrolyte ions.

## 1. Introduction

Hybrid materials have been extensively studied for energy storage applications due to the possibility of combining the advantages of two (or more) individual constituents to meet the demand for recent technological applications [1,2]. In recent years, coordination polymers (CPs), especially Prussian Blue (PB) and its analogues (PBAs), have been widely used for energy storage applications because of their unique features, such as their open frameworks, high stability, and redox activity. However, the poor conductivity of these materials presents a major obstacle for practical use. Hybrid materials composed of PB and rGO nanosheets have been investigated to some extent recently [2,3,4].

In general, carbon-based materials [5], such as graphene oxide (GO), reduced graphene oxide (rGO), and carbon nanotubes, or conducting polymers, such as polypyrrole (PPy) and polyaniline (PANI), are typically used to improve the conductivity of PB-based materials [6,7,8,9,10]. The active functional groups on the basal planes and edges of GO nanosheets allow for the adsorption of cations which act as nucleation centers for the growth of PB nanoparticles, thus making them attractive to be hybridized with PB. In addition, the developed reduced GO (rGO) nanosheets during composites synthesis possess high electrical conductivity and improved structural stability. Our group has reported the in-situ growth of various cyano-bridged CPs on the surface of GO by different methods [11].

PB and PBAs have been utilized as precursors for the preparation of metal oxides with good electrochemical performance for energy storage applications by means of thermal decomposition [12]. In particular, iron oxides (Fe*_x_*O*_y_*) have shown some promise as an electrode material for supercapacitors owing to their high theoretical specific capacitance, cost-effectiveness, low toxicity, excellent safety, and wide abundance [13,14]. In this work, we demonstrate the facile preparation of a PB/GO composite through the in-situ deposition of PB nanoparticles on GO nanosheets and the subsequent thermal treatment to convert the PB/GO composite to nanoporous Fe_2_O_3_/carbon composite (Scheme 1). The resulting Fe_2_O_3_/carbon composite shows high specific capacitance, excellent rate capability, and good cycling stability for supercapacitors. 

## 2. Experimental Section

### 2.1. Chemicals

Sodium ferrocyanide (II) decahydrate (Na_4_[Fe(CN)_6_]·10H_2_O) was purchased from Sigma-Aldrich, Co., St. Louis, MO, USA. The sulfuric acid solution was purchased from Nacalai tesque, Inc., Kyoto, Japan. Potassium hydroxide, sodium nitrate and iron (III) chloride hexahydrate (FeCl_3_·6H_2_O) were purchased from FUJIFILM Wako Pure Chemicals, Osaka, Japan. Nanographite platelets (N008-100-N) of 100 nm thickness were used as a raw material to prepare graphene oxide (GO) sheets (Angstron materials, Dayton, OH, USA). Potassium permanganate (KMnO_4_) and hydrogen peroxide (H_2_O_2_) were purchased from Kanto Chemicals Co., Inc., Tokyo, Japan. All chemical reagents were used without further purification.

### 2.2. Synthesis of GO Nanosheets

GO nanosheets were prepared according to the modified Hummer’s method [1]. In a typical process, graphite powder (N008-100-N, carbon source) and sodium nitrate were mixed together and then a concentrated sulphuric acid solution (7.67 mL) was added into this suspension under constant stirring for 1 h. Subsequently, KMnO_4_ (1.0 g) was added into the resulting mixture solution while maintaining the temperature at lower than 20 °C. The reaction mixture was then stirred at 35 °C for 2 h followed by the addition of pure water (83 mL) under vigorous stirring. To ensure the completion of the reaction, the obtained suspension was further treated with an aqueous H_2_O_2_ solution (30% *w*/*w*, 1.67 mL). The resulting solution was washed several times using the diluted HCl solution and distilled water. Finally, it was sonicated in pure water for several hours for exfoliation. The GO powder was then collected by centrifugation for 30 min at 14,500 rpm. The precipitate was washed again with distilled water for several times and dried naturally in air at room temperature and then, in vacuum at 60 °C overnight. The resulting GO powder was used to prepare the aqueous GO solution (2 mg mL^−1^) in the next step.

### 2.3. In-Situ Growth of PB Nanoparticles on the Surface of GO Nanosheets (PB/GO Composite) and the Subsequent Thermal Conversion to Nanoporous Iron Oxide/Carbon Composite

In a typical procedure, 40 mL of 0.299 mM FeCl_3_·6H_2_O solution was added dropwise to the GO solution (20 mL, 2 mg mL^−1^) followed by stirring for 30 min. The obtained mixture was gently mixed with a 0.358 mM Na_4_[Fe(CN)_6_]·10H_2_O solution (40 mL) by stirring for 30 min. The resulting suspension was aged for two days until the reaction was completed. The precipitate of the PB/GO composite was isolated by centrifugation (for 30 min at 14,500 rpm) and washed with water and ethanol several times. The precipitate was left to dry at room temperature for the next step. The nanoporous Fe_2_O_3_/carbon composite was obtained by heating the as-synthesized PB/GO powder at 350 °C for 1 h in air with a fixed heating rate of 5 °C min^−1^. The PB nanoparticles and GO nanosheets were also heated under the same conditions to compare their electrochemical performance with the Fe_2_O_3_/carbon composite. For the synthesis of PB nanoparticles, a 40 mL aqueous solution of 0.299 mM FeCl_3_·6H_2_O was mixed with another 40 mL aqueous solution of 0.358 mM Na_4_[Fe(CN)_6_]·10H_2_O and stirred for 1 h before aging overnight to ensure a complete reaction. The PB nanoparticles were collected by centrifugation (for 30 min at 14,500 rpm) and washed with water and ethanol several times. The precipitate dried naturally at room temperature (Appendix A).

### 2.4. Characterization

Wide-angle powder X-ray diffraction (PXRD) analysis was performed with Rigaku (Tokyo, Japan) RINT 2500X diffractometer using a monochromated Cu Kα (1.5406 Å, 40 kV, 40 mA) radiation. Nitrogen adsorption-desorption isotherms were obtained with a Quantachrome (Boynton Beach, FL, USA) Autosorb automated gas sorption system at 77 K. Before the measurement, all samples were dehydrated completely by heating at 250 °C overnight under vacuum. Morphological observations of the samples were performed using a scanning electron microscope (SEM, Hitachi SU8000, Tokyo, Japan) and transmission electron microscope (TEM, JEOL JEM2100, Tokyo, Japan). Fourier-transform infrared (FTIR) spectroscopy was performed using a Thermoscientific Nicolet 4700 (Waltham, MA, USA).

### 2.5. Electrochemical Measurements

All electrochemical measurements were carried out on an electrochemical analyzer (CH Instruments Model 600E, Austin, TX, USA). The electrochemical measurements of the prepared electrodes were investigated by the cyclic voltammetry (CV) and galvanostatic charge-discharge (GCD) using a three-electrode system with platinum wire as the counter electrode, Ag/AgCl electrode as the reference electrode, and 3 M KOH solution as the electrolyte. The working electrode was prepared by coating a slurry containing the active material (1 mg, 85 wt.%), carbon black (10 wt.%), polyvinylidene fluoride binder (PVDF) (5 wt.%), and N-methyl-2-pyrrolidone on a graphite paper (thickness: 1 mm) with an area of 1 × 1 cm^2^ as the current collector and dried at 60 °C. In our experiments, we used the flexible graphite paper as the current collector because of its excellent electrical conductivity as well as its negligible capacitive performance (usually below 1 F g^−1^), which would contribute little to the total capacitance of the prepared working electrode. The specific gravimetric capacitance of the electrode was calculated from the CV curves by using the following Equation (1):(1)$$C_{g}=\frac{1}{ms\left(Vf-Vi\right)}\overset{Vf}{\underset{Vi}{\int }}I\left(V\right)dv$$
where *C*_g_ is the gravimetric capacitance (F g^−1^), *s* is the potential scan rate (V s^−1^), *V* is the potential window (V), *I* is current (A), and *m* is the mass of active material (g).

Based on galvanostatic charge-discharge (GCD) curves, the gravimetric specific capacitance (*C*_g_, F g^−1^) was calculated using the Equation (2):(2)$$C_{g}=\;\frac{I\cdot t}{m\cdot \Delta V}\;$$
where *I* is the discharge current (A), *t* is the discharge time (s), *m* is the mass of active material (g), and Δ*V* is the potential change during the discharge process (V).

## 3. Results and Discussion

The functional groups, including hydroxyl, carboxyl and carbonyl on the GO nanosheets enables the adsorption of Fe^3+^ ions at the GO surface in the first step. Then, the adsorbed Fe^3+^ ions interact with the ferrocyanide ligand to initiate the growth of PB nanoparticles on the GO surface. The change in the surface charge of the GO nanosheet suspension before and after the addition of Fe^3+^ ions was studied by measuring the zeta potentials. The zeta potential measurements reveal that the surface charge of the GO nanosheets was changed from negative to positive due to the adsorbed cations (Fe^3+^ ions) [15,16]. During the growth of PB, the color of the suspension gradually changed to blue due to the formation of PB nanoparticles (Figure 1a). The morphology of the PB/GO composite was investigated by both SEM and TEM (Figure 1b,c). The high-magnification SEM and TEM images of the PB/GO composite clearly show the wrinkly structure of GO-wrapped PB nanoparticles.

The crystal structure and phase purity of the PB/GO composite were checked by XRD (Figure 2a). The XRD pattern of the GO nanosheets shows two sharp diffraction peaks at 2*θ* = 11° and 26° indexed to (001) and (002) planes of GO, respectively, indicating the highly ordered nature of the graphitic layers [17,18]. The XRD pattern of the PB nanoparticles (Figure 2a) is assignable to a typical face-centered cubic PB crystal structure (JCPDS No. 73-0687) [19]. The diffraction peaks of PB still remain in the PB/GO composite even after the hybridization with GO, however the characteristic peaks of the GO nanosheets are not detected because the anchored PB nanoparticles cover the surface of GO nanosheets, thus preventing their restacking.

The chemical compositions of the PB nanoparticles, GO nanosheets, and PB/GO composite were investigated by the FTIR spectroscopy (Figure 2b). The peaks observed in the IR spectrum of GO nanosheets can be assigned to the oxygen-containing functional groups on the GO surface [20,21,22]. After the hybridization with PB nanoparticles, several bands belonging to GO disappear and/or the peak intensities are largely reduced, due to the successful reduction of GO to reduced graphene oxide (rGO). The peak at 2080 cm^−1^ is attributed to –C≡N– units, thus confirming the existence of PB nanoparticles [23,24]. The peak at 1600 cm^−1^ on the PB/GO composite is difficult to identify as it overlaps with the C=C group of GO nanosheets and the –OH group of PB nanoparticles.

As shown in Scheme 1, the thermal conversion of the PB/GO composite at 350 °C in air results in the formation of the iron oxide (Fe_2_O_3_)/carbon composite [25]. The crystal structure of the PB/GO composite after the heat treatment was examined by the wide-angle XRD (Figure 3a). The original peaks of PB vanished entirely. The new and broad peaks at 2θ = 35° and 63° can be assigned to impurity-free γ-Fe_2_O_3_ [26], thereby confirming the successful conversion of PB to iron oxide. The diffraction peaks of GO nanosheets are not observed in the prepared Fe_2_O_3_/carbon composite. The morphology of this composite was investigated using both SEM and TEM (Figure 3b–d). As revealed by the SEM image, the size of the resulting iron oxide nanoparticles is nearly similar as that of the starting PB nanoparticles (Figure 1b). The high-resolution SEM image shows that the iron oxide nanoparticles are aggregated to each other. To further investigate the morphology of this composite, TEM analysis was carried out. Figure 3c confirms the presence of nanoparticle aggregates and wrinkled GO nanosheets. The high-resolution TEM (HRTEM) image shows several lattice fringes belonging to iron oxide (Figure 3d). The EDX elemental mapping shows the uniform distribution of carbon (C), oxygen (O), and iron (Fe) atoms (Figure 4). The Fe:C ratio of the resulting iron oxide/carbon composite is around 62:38 (wt.%). The presence of Fe and C was also confirmed by FTIR spectroscopy, as shown in Appendix A.

Figure 5 shows the N_2_ adsorption-desorption isotherms of PB nanoparticles and PB/GO composite before and after thermal treatment. The textural characteristics are summarized in Table 1. The Brunauer-Emmett-Teller (BET) specific surface area of the PB/GO composite (152.6 m^2^ g^−1^) is remarkably improved compared to pure GO (34.9 m^2^ g^−1^) [1] and pure PB (36.1 m^2^ g^−1^). The adsorption isotherms are gradually increased with the increase of relative pressure, indicating that various pores with different pore sizes are randomly distributed over the entire area. The higher surface area of the PB/GO composite can be attributed to the good dispersion of the PB nanoparticles on the GO surface, thus enabling them to act as spacers to prevent the GO layers from aggregation or restacking. Following the heat treatment, the surface area of the iron oxide/carbon composite is mostly similar to that of the precursor PB/GO composite.

It is well-reported that graphene/metal oxide composites exhibit good electrochemical performance for supercapacitor applications [27,28]. The electrochemical properties of pure GO, pure iron oxide, and iron oxide/carbon composite electrodes for supercapacitor applications were investigated by using a three-electrode system in a 3 M KOH electrolyte. The cyclic voltammetry (CV) curves of the pure GO electrode at different scan rates (Figure 6a) reveal rectangle-like curves, indicating its electrical double layer capacitive properties, while in the case of iron oxide and iron oxide/carbon electrodes (Figure 6b,c) the CV curves exhibit a pair of redox peaks due to pseudocapacitive behavior. The existence of an oxidation peak at ca. 0.3 V and a reduction peak at ca. −0.7 V vs. Ag/AgCl, [29] for the iron oxide/carbon composite electrode implies reversible redox reactions of iron ion [30].

Based on CV measurements, the specific capacitance values of GO, iron oxide, and iron oxide/carbon electrodes at a scan rate of 2 mV s^−1^ are 220.0, 264.0, and 551.5 F g^−1^, respectively. More importantly, the iron oxide/carbon composite electrode exhibits higher specific capacitance values than both GO and iron oxide electrodes at all scan rates owing to the synergistic combination of EDLC from GO and pseudocapacitance from Fe_2_O_3_ (Table 2). The capacitance retention value of the iron oxide/carbon composite electrode at 100 mV s^−1^ is very high (50%), superior to those of graphene oxide (31%) and iron oxide electrodes (44%).

The long-term stability of the iron oxide/carbon composite electrode was investigated by the CV measurement at 100 mV s^−1^ (Figure 6d). Clearly, the capacitance gradually increases until the capacitance retention reaches 120% after 8000 cycles. The increase in the specific capacitance during cycling is possibly due to the activation of the hybrid electrode material as a result of the improved surface wetting of the electrode, leading to improvement in the electrolyte ion diffusion. Furthermore, the as-prepared iron oxide/carbon composite electrode shows better electrochemical performance for supercapacitor applications than previously reported iron oxide/GO electrodes in terms of specific capacitance and capacitance retention (Appendix A).

Galvanostatic charge-discharge (GCD) measurements were carried out to investigate the rate performance of the graphene oxide, iron oxide, and iron oxide/carbon electrodes at current densities of 2, 4, 6, 8, 10, and 20 A g^−1^ (Figure 7). The GCD curves of the iron oxide/carbon composite electrode depicts high specific capacitance in comparison with graphene oxide and iron oxide. A non-linear behavior was observed during charging (at a potential of ca. 0.3 V) and discharging (at a potential of ca. −0.7 V) vs. Ag/AgCl and these observations are in good agreement with the oxidation and reduction peaks observed in the CV curves. Figure 7d compares the rate capabilities of graphene oxide, iron oxide, and iron oxide/carbon composite electrodes at different current densities. At a current density of 2 A g^−1^, the specific capacitance of the iron oxide/carbon composite electrode is 415.0 F g^−1^, which is much higher than those of graphene oxide (210.0 F g^−1^) and iron oxide electrodes (202.5 F g^−1^). Notably, with the increase of current density to 20 A g^−1^, the iron oxide/carbon composite electrode still exhibits a high capacitance retention of 51%. On the contrary, pure graphene oxide and iron oxide electrodes display poor capacitance retention of 21% and 29%, respectively, indicating that the hybrid structure was beneficial for enhancing the diffusion of electrolyte ions.

## 4. Conclusions

In this study, PB nanoparticles were simultaneously deposited on the surface of GO nanosheets through the adsorption of positively charged Fe^3+^ ions on GO surface firstly followed by addition of Na_4_[Fe(CN)_6_]·10H_2_O. The oxygen-containing functional groups of the GO surface served as excellent anchoring agents for the growth of PB nanoparticles. The PB/GO composite precursor was successfully converted into the iron oxide/GO composite with a well-retained morphology by a facile thermal annealing in air at 350 °C. The obtained iron oxide/carbon composite shows superior electrochemical performance as an electrode material for supercapacitors compared to both pure iron oxide and graphene oxide electrodes in terms of specific capacitance, rate capability, and capacitance retention. The results will provide a useful guidance for constructing high-performance hybrid materials for energy storage applications using PB and PBA.

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
