# Peer review of "Nanoporous Iron Oxide/Carbon Composites through In-Situ Deposition of Prussian Blue Nanoparticles on Graphene Oxide Nanosheets and Subsequent Thermal Treatment for Supercapacitor Applications"

_nanomaterials, 2019, doi:10.3390/nano9050776_

Round 1
Reviewer 1 Report
The article describes the preparation of graphene/iron oxide hybrid starting from PB. The article also presents the electrochemical performance of a supercapacitor device fabricated from the rGO/PB hybrid. The results worth publishing after considering the following comments
The title should be changed to reflect the content. The title should particularly referee to the supercapacitor device
The abstract is too short and need to include some more information about the method used in the synthesis and the results of the supercapacitor.
Introduction:
The chemical formula of PB should be introduced here (or better in the abstract)
CNTs have been used before to grow nanoparticles on it for several applications (including SC). The authors should include this discussion in the introduction. See for example DOI: 10.1002/adma.201003658, DOI: 10.1016/j.elecom.2008.08.051, DOI: 10.1088/1361-6528/aae5c6, DOI: 10.1039/c2ee23284g
Results and discussion:
The conversion of PB to iron oxide is always associated with releasing gases. This usually introduces some porosity to the structure, which is not shown by the nitrogen gas adsorption/desorption isotherm in figure 5. The authors should comment on that
Why the particles size of PB before and after heat treatment was the same despite the change in the chemical composition and phases?
Author Response
**All the revised parts are highlighted in yellow color.**
1-1: The title should be changed to reflect the content. The title should particularly referee to the supercapacitor device. The abstract is too short and need to include some more information about the method used in the synthesis and the results of the supercapacitor.
Reply: According to the comments, we revised the title and the abstract. Please see the title and the abstract.
1-2: The chemical formula of PB should be introduced here (or better in the abstract)
Reply: The chemical formula of PB is a little complicated. It is changed, depending on the synthetic conditions. So, in this paper, we would like to mention PB simply.
1-3: CNTs have been used before to grow nanoparticles on it for several applications (including SC). The authors should include this discussion in the introduction. See for example DOI: 10.1002/adma.201003658, DOI: 10.1016/j.elecom.2008.08.051, DOI: 10.1088/1361-6528/aae5c6, DOI: 10.1039/c2ee23284g
Reply: Thanks. We cited these references.
1-4: The conversion of PB to iron oxide is always associated with releasing gases. This usually introduces some porosity to the structure, which is not shown by the nitrogen gas adsorption/desorption isotherm in Figure 5. The authors should comment on that.
Reply: Yes, thanks. The adsorption isotherms are gradually increased with the increase of relative pressure, indicating that various pores with different pore sizes are randomly distributed over the entire area. Please see Page 7.
1-5: Why the particles size of PB before and after heat treatment was the same despite the change in the chemical composition and phases?
Reply: Yes, this is common phenomenon. If the particles are seriously shrunk, they cannot keep the porous structures inside the particles.
Reviewer 2 Report
This study defins a good and interested starting point. However supercap performance, even if better than for compared systems, is still far away from the ideal behaviour and should be improved in the future if authors wish their invention to be widely used.
Author Response
**All the revised parts are highlighted in yellow color.**
2-1: This study defines a good and interested starting point. However, supercapacitor performance, even if better than for compared systems, is still far away from the ideal behavior and should be improved in the future if authors wish their invention to be widely used.
Reply: Thanks for high evaluation of our manuscript. Yes, we will try to demonstrate the real device application in the next paper.
Reviewer 3 Report
The manuscript by A. Azhar and co-workers reported iron oxide/carbon based hybrid materials through thermal treatment of Prussian blue deposited onto graphene oxide. The resulting hybrid material is demonstrated as a promising material for supercapacitor. However, the novelty of this work is not clear. There is a lot of work published on iron oxide/carbon and/or graphene based composites for supercapacitor applications (for example, Chem. Engr. J. 2016, 286, 165-173; Nano Energy 2014, 7, 86-96; Adv. Mater. 2011, 23, 5574-5580 etc.) and some of these works achieved significantly higher specific capacitance compared to the capacitance reported in this manuscript. The authors did not discussed about the previous works in the manuscript. Moreover, S. Tanaka et al. (RSC Adv. 2017, 7, 33994-33999) reported on Prussian blue derived iron oxide nanoparticles wrapped in graphene oxide for supercapacitors by thermal annealing at 400 C in air which is quite similar to the material preparation in this work. Therefore, the manuscript is not suitable for publication in this journal.
Additional comments:
1. There are a lot of spelling mistakes throughout the manuscript, for example, in page-1, line-38 ‘’grphene and carbon nanotcubes’’; line-42 ‘’edeges’’; line-43 ‘’necleation’’; page-5, line-173 ‘’charaxterustic’’ etc. Authors need to check the manuscript carefully before submitting to a journal.
2. Did authors used Hummers method for the synthesis of graphene oxide? Also the purification step of the synthesized GO should be mentioned in the experimental section.
3. What was the centrifugation rate and time used to isolate PB/GO hybrids?
4. The text in Figure 3d inset cannot be seen clearly. Authors need to change the colour of the text.
5. This work achieved a specific capacitance of 551.5 F/g at a scan rate of 2 mV/s from iron oxide/carbon composite which needs to be compared with previous works on iron oxide/carbon based materials.
6. What is the power density and energy density of the fabricated supercapacitors?
Author Response
**All the revised parts are highlighted in yellow color.**
3-1: There are a lot of spelling mistakes throughout the manuscript, for example, in page-1, line-38 “grphene and carbon nanotcubes’’; line-42 ‘’edeges’’; line-43 ‘’necleation’’; page-5, line-173 ‘’charaxterustic’’ etc. Authors need to check the manuscript carefully before submitting to a journal.
Reply: So sorry for this. When the file was converted to the PDF file, this problem happened. Anyhow, in the revision stage, we carefully uploaded the files without any careless mistakes.
3-2: Did authors used Hummers method for the synthesis of graphene oxide? Also, the purification step of the synthesized GO should be mentioned in the experimental section. What was the centrifugation rate and time used to isolate PB/GO hybrids?
Reply: We have already mentioned this point briefly in the manuscript. Anyhow, according to the comments, we extended the sentences in the experimental section. Please see Page 3-4.
3-3: The text in Figure 3d inset cannot be seen clearly. Authors need to change the color of the text.
Reply: According to the comment, we revised the figure. The inset figure of Figure 3d is not necessary, so we deleted this part.
3-4: This work achieved a specific capacitance of 551.5 F/g at a scan rate of 2 mV/s from iron oxide/carbon composite which needs to be compared with previous works on iron oxide/carbon based materials.
Reply: We have already addressed this point. Please see Table S1.
3-5: What is the power density and energy density of the fabricated supercapacitors?
Reply: This paper is focusing on the materials synthesis. So, we would like to show only three electrode system for calculating the capacitance.
Round 2
Reviewer 3 Report
The paper presented by Azhar and co-workers has
been improved compared to the prior version and I recommended it for publication once the following comment is addressed:
In page-2, line 47, the authors mentioned ''GO nanosheet possess high electrical conductivity and improved structural stability'' which is incorrect because the electrical conductivity of GO is very poor and basically it's an insulating material. To make it electrically conductive reduction of oxygen function groups are necessary. Therefore, I advise the authors to correct the sentence and replace GO with ''reduced GO'' or ''graphene''
Author Response
Yes, we used "reduced GO", according to the comments.